# Fighting Non-Small Lung Cancer Cells Using Optimal Functionalization of Targeted Carbon Quantum Dots Derived from Natural Sources Might Provide Potential Therapeutic and Cancer Bio Image Strategies

**DOI:** 10.3390/ijms232113283

**Published:** 2022-10-31

**Authors:** Hanaa Mohammed Elsayed Mohammed El-brolsy, Nemany A. N. Hanafy, Maged A. El-Kemary

**Affiliations:** Nanomedicine Group, Institute of Nanoscience and Nanotechnology, Kafrelsheikh University, Kafrelsheikh 33516, Egypt

**Keywords:** carbon quantum dots, cytotoxicity, medical bio-image, nuclear condensation

## Abstract

Non-small cell lung cancer (NSCLC) is an important sub-type of lung cancer associated with poor diagnosis and therapy. Innovative multi-functional systems are urgently needed to overcome the invasiveness of NSCLC. Carbon quantum dots (CQDs) derived from natural sources have received interest for their potential in medical bio-imaging due to their unique properties, which are characterized by their water solubility, biocompatibility, simple synthesis, and low cytotoxicity. In the current study, ethylene-diamine doped CQDs enhanced their cytotoxicity (98 ± 0.4%, 97 ± 0.38%, 95.8 ± 0.15%, 86 ± 0.15%, 12.5 ± 0.14%) compared to CQDs alone (99 ± 0.2%, 98 ± 1.7%, 96 ± 0.8%, 93 ± 0.38%, 91 ± 1.3%) at serial concentrations (0.1, 1, 10, 100, 1000 μg/mL). In order to increase their location in a specific tumor site, folic acid was used to raise their functional folate recognition. The apoptotic feature of A549 lung cells exposed to N-CQDs and FA-NCQDs was characterized by a light orange-red color under fluorescence microscopy. Additionally, much nuclear fragmentation and condensation were seen. Flow cytometry results showed that the percentage of cells in late apoptosis and necrosis increased significantly in treated cells to (19.7 ± 0.03%), (27.6 ± 0.06%) compared to untreated cells (4.6 ± 0.02%), (3.5 ± 0.02%), respectively. Additionally, cell cycle arrest showed a strong reduction in cell numbers in the S phase (14 ± 0.9%) compared to untreated cells (29 ± 0.5%). Caspase-3 levels were increased significantly in A549 exposed to N-CQDs (2.67 ± 0.2 ng/mL) and FA-NCQDs (3.43 ± 0.05 ng/mL) compared to untreated cells (0.34 ± 0.04 ng/mL). The functionalization of CQDs derived from natural sources has proven their potential application to fight off non-small lung cancer.

## 1. Introduction

Lung cancer is the second most famous form of cancer in both men and women because it is responsible for a significant number of cancer-related deaths around the world. It reported that 85% of lung cancer-related deaths are attributed to non-small cell lung cancer (NSCLC) [1]. This type of cancer has the ability to metastasize into the brain, bones, and liver. This behavior highlights the high unmet medical need for the treatment of advanced lung cancer [2]. Cisplatin is a popular chemotherapeutic agent used to treat lung cancer. However, it causes adverse effects, including hepatotoxicity, nephrotoxicity, and cardiotoxicity, and is associated with disorders in the antioxidant defense system, cardiomyopathy, arrhythmias, and myocarditis [3]. Chemotherapies, such as paclitaxel, carboplatin, and doxorubicin, are important strategies in cancer treatment. However, they have a number of disadvantages, including poor selectivity, poor adsorption, strong drug resistance, and negative side effects [4]. Additionally, creating chemotherapeutic drugs capable of fighting specific cancer cells without causing any damage to normal cells is still a difficult task in cancer treatment [5]. Additionally, health care for cancer patients is affected by the cost-effectiveness of chemotherapies. 

Today, nanoparticles made from natural sources are used to develop effective and affordable strategies for the early detection, diagnosis, and treatment of cancers [6]. Early diagnosis enables timelier treatment, significantly improves patient outcomes, and is essential for successful therapy [7]. Carbon-based quantum dots (CDs) are mainly divided into two subgroups, carbon quantum dots (CQDs) and graphene quantum dots (GQDs), and have gained extensive attention in scientific areas in recent years. CQDs were first reported in 2004 by purification of single-walled carbon nanotubes [8]. Meanwhile, CQDs derived from natural sources have recently received much attention for a variety of applications because of their biocompatible and long-lasting fluorophores [9,10,11].

In modern nanotechnology, CQDs have been used to fabricate drug delivery, cellular bioimage, bio-sensing, and other biomedical applications [12,13].

CQD nanostructures are synthesized using various techniques, including microwave irradiation, electrochemical oxidation, hydrothermal method, laser ablation, reflux method, ultrasonication, pyrolysis of glycerol, strong acidic and electrochemical oxidation, thermal carbonization of molecules, thermal annealing of barbecue meat (BBQ) char, as well as atmospheric plasma-based synthesis [14,15]. However, this type of synthesis needs many chemical reactions, such as oxidation, pyrolysis, and carbonization. Therefore, green synthesis of CQDs has several advantages, such as renewable resources, the use of low-cost and non-toxic raw materials, simple operations, and being environmentally-friendly [16].

According to the previous literature, many natural materials have been used as green sources and precursors to produce CQDs, such as jackfruit, apple, orange, cabbage, banana, pear, jujubes, oolong tea, betel leaf, bamboo leaf, and guava leaf. These materials exhibited outstanding properties, including high renewable capability, low cost, high yield, high availability, and high biocompatibility [17]. A report was published in 2020 using chicken blood as a low-cost material. In this report, CQDs were prepared from chicken blood using a hydrothermal method that used a biosensor for catalyzing the oxidation of 3, 3′, 5, 5′-tetramethylbenzidine (TMB) in the presence of H2O2 to generate blue oxidized TMB (ox-TMB) with a strong absorption peak at 652 nm [18].

In the current study, CQDs were synthesized in a facile and economic synthetic approach from a biological source (Chicken Blood; CB) due to its low cost and availability containing a multi-element mixture, followed by doping CB-CQDs with a nitrogen source using ethylene-diamine (EDA) to enhance the chemical–physical properties of CQDs with a hydrothermal carbonization method [19]. This functionalization was termed nitrogen-doped CQDs (N-CQDs). Additionally, their surface can be further modified to improve the functionality of nanoparticles for specific targeting therapies in biomedical and pharmaceutical applications. For example, folic acid (FA), a biological micro-molecule that plays an important role in various biological functions, has been extensively used to modify nanoparticles, in particular, for cancer-targeting therapies [20]. Folic acid-functionalized NCQDs (FA-NCQDs) can be internalized and accumulated in the prenuclear region of cytoplasm as a result of the defensive function of endosomes or lysosomes [21].

In our previous work, FA-conjugated NPs derived the encapsulated cargo molecules into specific cancer locations, allowing them to increase their cellular uptake. This folic acid–folate receptors conjugation was used as a mechanism to facilitate cancer delivery and was studied extensively in breast cancer, lung cancer, and hepatocellular carcinoma [22].

In the current study, non-small lung cancer exposed to FA-NCQDs exhibited activation of the proapoptotic pathway. This result suggests that these functionalized nano-delivery systems might be promising materials for further applications in in vitro bio-imaging and can be used for targeted cancer therapy because of their low cost compared to standard chemotherapies and their availability, easy fabrication, and functionalization.

## 2. Results

### 2.1. Characterization

In the current study, the amount of CQDs obtained from 10 gm of chicken blood after purification and lyophilization was 500 mg, while the amount of N-CQDs obtained from 2gm chicken blood and 2ml ethylene diamine was 1300 mg.

N-doped CQDs (N-CQDs) exhibited absorption peaks in the UV region at 270 nm, which was aroused from π-π* transition of the C=C bonds, and another distant peak at around 331 nm, which was aroused from n-π* transition of the C=N bonds [23,24,25]. In the recent study, the N-CQDs conjugated FA through interaction between carboxyl and amino groups. The alkaline solution is used to deprotonate carboxyl groups of FA and then facilitates their further interaction. The characteristic peak of FA-NCQDs measured at 288 nm was assigned to π-π* transition of C=C bonds. Meanwhile, another visible transition of C=O appeared at 365 nm representing FA was clearly observed in the UV–vis absorption of N-CQDs. This confirms the successful conjugation of FA onto the surface of N-CQDs (Figure 1) [26,27,28,29].

In the recent work, N-CQDs showed a low negative zeta potential value (−17 mv) compared to CB-CQDs(−24 mV) due to the presence of amino groups of EDA [30,31,32,33]. Meanwhile, the zeta potential of pure FA obtained −27 mV. While the zeta potential for FA functionalized, N-CQDs changed into −10 mV [34].

N-CQDs appeared as spherical shapes in the TEM image and were individually distributed without any aggregation. Their size ranged from 3.7 to 6.7 nm, with an average diameter of about 4.9 nm (Figure 2) [23,24].

Folic acid has a natural crystallinity structure showing multiple peaks at angles of (2θ = 10°, 12°, 16°, 21°, 25°, 29°, 37°, 43°, 64°, 77°). The XRD spectrum of FA-NCQDs shows one peak at (2θ = 23.4°) that is attributed to the peak of CB-CQDs and N-CQDs. Additionally, the other peaks of folic acid were shifted to angles of (2θ = 14°, 27°, 28°, 31°, 32°, 35°, 38°, 45°, 65°, and 74°) (Figure 3A) [29].

The FTIR spectrum of FA shows a broad N−H/ O-H stretching peak at 3325 cm^−1^ along with C−H vibrations at 2819 cm^−1^, C=O stretching vibration at 1712 cm^−1^, C=C unsaturated bonds at 1610 cm^−1^ and C−NH stretching vibration at 1520 cm^−1^. In the FA-NCQDs, the OH /N−H stretching shifted to 3264 cm^−1^, while the stretching vibrations of C=O and C−NH shifted to 1639 and 1560 cm^−1^, respectively. Moreover, the appearance of the aromatic or aliphatic C−H vibrations of FA at 2810 cm^−1^ confirms that N-CQDs were being functionalized by the presence of FA (Figure 3B) [29].

### 2.2. Cellular Experiments (In Vitro Analysis)

#### 2.2.1. Cytotoxicity Assay

Cytotoxicity assays are extensively used in pharmaceutical research, drug discovery, and other fundamental research to evaluate the toxicity of molecules. In the current study, Sulforhodamine B (SRB), a very small aminoxanthene molecule that can penetrate cell plasma membrane into the cytoplasm and reacts with a basic amino acid of protein in the presence of acetic condition forming aggregated SRB-protein complex, was used. Here, the SRB reagent was used to measure spectrophotometry of the proliferation of the A549 cell line of non-small lung cancer after their incubation to serial concentrations (0.1, 1, 10, 100, 1000 μg/mL) of CB-CQDs, EDA, N-CQDs, and FA-NCQDs. The growth of non-small lung cancer (A549) cell lines showed significant reduction after their exposure to EDA (99 ± 0.03%, 97 ± 0.8%, 94 ± 1.5%, 48 ± 0.8%, 1.5 ± 0.2%), respectively, and N-CQDs (98, ± 0.4%, 97 ± 0.38%, 95.8 ± 0.15%, 86 ± 0.15%, 12.5 ± 0.14%), respectively, FA-NCQDs (96 ± 0.3%, 94 ± 0.5%, 93 ± 1.5%, 83 ± 1.2%, 10 ± 0.8%), respectively, compared to CB-CQDs (99 ± 0.2%, 98 ± 1.7%, 96 ± 0.8%, 93 ± 0.38%, 91 ± 1.3%), respectively (Figure 4). Additionally, the potential inhibitory effect was seen after their incubation to concentration (1000 µg) exhibiting cytotoxic activity (1.5 ± 0.2%) in case EDA and (12.5 ± 0.14%) and (10 ± 0.8%) in case N-CQDs and FA-NCQDs, respectively. This confirms that the toxicity of N-CQDs and FA-NCQDs is due to the presence of EDA, which was used as a source of nitrogen. To confirm the potential therapeutic effect of N-CQDs and to use them later on an animal cancer model, the half-maximal inhibitory concentration (IC50) values for both CB-CQDs, EDA, N-CQDs, and FA-CQDs were detected by using serial concentrations (0.1, 1, 10, 100, 1000 μg/mL) (Figure 4). The result showed that the cytotoxic activity of CB-CQDs EDA, N-CQDs, and FA-NCQDs was detected at (IC50 > 1000 µg/mL, IC50: 95 µg/mL, IC50:304 µg/mL, IC50: 210 µg/mL), respectively [35,36] (Figure 5).

Normal Human Skin Fibroblast cell lines (NHSF) were exposed to serial concentrations (0.1, 1, 10, 100, 1000 μg/mL) of CB-CQDs, N-CQDs, and FA-NCQDs. The result was measured through spectrophotometry using SRB reagent after 72 h. The growth of NHSF cells showed significant reduction after their exposure to N-CQDs (101 ± 1.7%, 96 ± 0.6%, 95 ± 1.3%, 83 ± 1.4%, 38 ± 0.7%), respectively, compared to CB-CQDs (97 ± 0.7%, 95 ± 1.7%, 94 ± 1.5%, 94 ± 0.9%, 93 ± 2.4%), while, the result of FA-NCQDs showed (101 ± 1.2%, 98 ± 0.6%, 95 ± 0.7%, 92 ± 0.3%, 90 ± 0.9%), respectively (Figure 6).

The potential inhibitory effect of concentration (1000 µg) exhibited (38 ± 0.7%) in the case N-CQDs, while less reduction was obtained (90 ± 0.9%) and (93 ± 2.4%) in the case FA-NCQDs and CB-CQDs, respectively. Thereby, IC50 of CB-CQDs, N-CQDs, and FA-NCQDs were detected at (IC50 > 1000 µg/mL, IC50: 612.15 µg/mL, IC50 > 1000 µg/mL), respectively (Figure 7). The result confirms that FA-functionalized N-CQDs can improve their delivery for cancer cells and reduce their toxicity on normal cells.

#### 2.2.2. Fluorescence Imaging of A549 Tumor Cells

The unique properties of CQDs, including tunable fluorescence and size below 10 nm, strongly recommend their use in biomedical imaging. The cellular uptake of CB-CQDs, N-CQDs, and FA-NCQDs was detected after their incubation for 24 h. A549 cells exposed to these materials exhibited fluorescence emission at blue, red, and yellow channels. Additionally, A549 cells exposed to folic acid conjugated N-CQDs showed good emission because folic acid might increase their cellular adsorption and facilitate their accumulation inside cytoplasm due to its functional attachment with folate receptors (Figure 8) [37]. Fluorescence intensity was measured using the Image J program, and the corrected total cell fluorescence (CTCF) was detected. The fluorescence emission has good intensity in cells exposed to FA-NCQDs. This is mainly attributed to accumulated nanoparticles in the perinuclear region of cancer cells. This accumulation is due to the presence of FA, which facilitates their cellular uptake (Figure 8A–C).

### 2.3. Cellular Morphological Alterations

The morphological alterations of non-small lung cancer (A549 cell line) exposed to CB-CQDs, N-CQDs, and FA-NCQDs were studied using Acridine orange/Ethidium bromide (AO/EtBr), DAPI, Crystal violet, and Propidium iodide stains [38].

#### 2.3.1. AO/EtBr

A549 cells (untreated cells) appeared in a well-organized structure showing green color under fluorescence microscopy, while early apoptotic cells were seen as light orange, and late apoptotic cells were shown as red fluorescence in the treated A549 cells. Additionally, many cells had undergone necrosis and cell degradation. For this reason, the number of cells in the section was reduced, particularly in the case of N-CQDs and FA-CQDs. These findings revealed that N-CQDs and FA-NCQDs have the ability to cause apoptosis and cell death in A549 cells (Figure 9) [39,40,41,42].

#### 2.3.2. DAPI Staining

The condensed and fragmented chromatins were investigated by using DAPI stain. The exposure of A549 cells to CB-CQDs, N-CQDs, and FA-NCQDs resulted in an increase in chromatin condensation and fragmentation compared to untreated cells (control). These results revealed that N-CQDs and FA-NCQDs caused high damage to the chromatin structure compared to CB-CQDs, which have low toxicity. Additionally, the quantification analysis showed an increase in the number of cells associated with nuclear fragmentation, nuclear condensation, nuclear degradation, and rupture of the cell membrane (Figure 10 and Figure 11) [43].

#### 2.3.3. Crystal Violet Stain

A549 cells exposed to CB-CQDs, N-CQDs, and FA-NCQDs (300 µL, or 800 µL per well/1mg/mL) for 24, 48, and 72 h exhibited a significant increase in the number of dead cells, while there were no dead cells detected in untreated cells (control). This result confirmed the potential cytotoxic activity of N-CQDs and FA-NCQDs against A549 compared to CB-CQDs that obtained less cytotoxicity (Figure 12 and Figure 13) [44].

#### 2.3.4. Propidium Iodide Staining

Propidium Iodide (PI) is a versatile dye. It can be used as an indicator for detecting dead cells among many non-damaged cells by the mechanism of its interaction with DNA and emitting orange-red fluorescence. PI cannot penetrate the cytoplasm of non-damaged cells. This mechanism was used to distinguish apoptotic cells among many non-damaged cells. In the current study, untreated cells (control) showed green fluorescence emission, while A594 cells exposed to N–CQDs and FA-NCQDs for 48 h exhibited red fluorescence. Indeed, the integral membrane structure of A594 cells can be damaged by the reaction of CB-CQDs, N-CQDs, and FA-NCQDs, leading to forming of a permeable membrane. Thereby, PI can be diffused through the membrane and react with DNA. After its binding to DNA, the quantum yield of PI was duplicated, and the emission shifted from green to orange-red. For this reason, PI was used extensively in flow cytometry to evaluate cell viability and cell cycle arrest and also used in fluorescence microscopy to differentiate apoptotic cells. (Figure 14) [45,46,47,48].

### 2.4. Flow Cytometry Results

Flow cytometry or fluorescence-activated cell sorting (FACS) is a standard technique used to measure cell apoptosis with high sensitivity and cell cycle analysis. The non-small lung cancer cells (A549 cell line) were exposed to 300 µL of 1 mg/1 mL N-CQDs and then used to investigate the percentage of apoptotic, necrotic cells and cell cycle arrest (Figure 15 and Figure 16) [49].

#### 2.4.1. Annexin V-FITC Apoptosis Detection Assay

The result of flow cytometry suggests a distinct increase in necrosis and late apoptotic stages after their incubation with N-CQDs compared to control cells, as shown in Figure 14. The percentage of cells in the late apoptosis was increased significantly in treated cells to (19.7 ± 0.03%, *p* < 0.05), compared to untreated cells (4.6 ± 0.02%). Similarly, necrosis was also increased in treated cells to (27.6 ± 0.06%, *p* < 0.05) compared to untreated cells (3.5 ± 0.02%). Total cell death was increased significantly in A549 cells after their treatment by N-CQDs to (48.2 ± 0.06%, *p* < 0.05) compared to untreated cells (control) that were obtained (11.3 ± 0.04%) [50].

#### 2.4.2. Cell Cycle Arrest

The cell cycle arrest of both untreated (control) and treated A549 cells were analyzed and classified into sub-G1, G1, S, and G2/M phases. In the control group (untreated A549 cells), the distribution of cells in the G1, S, and G2 phases of the cell cycle were estimated as 60 ± 0.8%, 29 ± 0.5%, and 10 ± 0.3%, respectively. In contrast, A549 cells exposed to N-CQDs for 48 h resulted in an increase in the number of cells in the G1 phase (70 ± 0.5%) with a strong reduction of cell numbers in the S phase (14 ± 0.9%, *p* < 0.001) compared to control (Figure 14) [51].

Figure 14C represents the percentage of cells in different cell cycle stages (G1, S, and G2) compared to control. In this diagram, the percentage of increased or decreased levels for each stage can be identified qualitatively and quantitatively compared to the same stage in control. For this reason, three stages (G1, S, and G2) were presented. Figure 15D represents the significant reduction in the percentage of cells for just the S phase compared to the same stage in control.

#### 2.4.3. ELISA Caspase Detection

Caspase-3 is a cysteine–aspartic acid protease that cleaves cellular targets and executes cell death. The role of caspase-3 in apoptosis is to cleave and activate caspases −6, −7, and −9 in order to break down the apoptotic cells before removal. The overexpression of caspase-3 enhances the chemo-sensitivity against the acquired drug resistance. In the current study, the caspase-3 level was increased significantly in A549 exposed to N-CQDs (2.67 ± 0.2 ng/mL, *p* < 0.01) and FA-NCQDs (3.43 ± 0.05 ng/mL, *p* < 0.001) compared to untreated cells (0.34 ± 0.04 ng/mL). Thereafter, the activated cytoplasmic caspase can contribute to the execution of apoptosis (Figure 17) [52].

## 3. Discussion

For the first time, the current study investigated the potential therapeutic effect of CQDs derived from chicken blood because of their content from a natural source of carbon and nitrogen. As an advantage, CQDs can be used in the future as alternative materials instead of chemotherapies because of their unique properties, including good solubility, high stability, biocompatibility, easy preparation, low cost, and low cytotoxicity. The next advantage is that their surface can be functionalized by EDA (as a source of nitrogen) and FA (for being used as targeted therapies) [53,54]. It was previously reported that FA-conjugated QDs results in oxidation of their surface. This could change the optical chemistry of QDs [55]. In a recent study, CQD functionalized by ethylene diamine was first obtained, and then FA dissolved by 1N sodium hydroxide was added. The alkaline solution was used here to deprotonate the carboxyl group, and then the (COO^−^) was activated and attached amino group of ethylene diamine. Similarly, in previous work, FA was first conjugated to chitosan, and the conjugation was used to encapsulate QDs. This modification improved the optical chemistry of QDs [56].

Here, two peaks of CQDs and N-CQDs were measured at 270 nm and 331 nm, respectively, owing to π–π* transition related to C=C and C=N bonds [23,24,25], while characteristic peaks of FA appeared at 288 nm and 365 nm in spectrum of FA-NCQDs (Figure 1), confirming their successful passivation and functionalization methods.

FTIR was used to identify the chemical band distribution of CQDs, N-CQDs, and FA-NCQDs. FTIR spectra showed that the characteristic peaks at 3420 cm^−1^, 3410 cm^−1^, and 3325 cm^−1^, respectively, can be attributed to the O–H/N–H stretching vibration of hydroxyl and/ or amino groups, while band C=O was located at 1667 cm^−1^, 1650 cm^−1^, and 1639 cm^−1^, in the spectra of CQDs, N-CQDs, and FA-NCQDs, respectively. The result confirmed the presence of characteristic peaks of CQDs. Meanwhile, their shape and diameter were detected using TEM. They appeared as spherical spots distributed at uniform dispersion without any aggregation. (Figure 2) and exhibited a size range of 3.7–6.7 nm with an average diameter of about 4.9 nm. This narrow diameter is mostly related to the hydrothermal method used for fabrication [18].

To explore the crystallite formation in CQD moieties, the lattice parameters of the CQDs, N-CQDs, and FA-NCQDs materials were observed with the X-ray diffraction (XRD) pattern. The result confirms that the structure of crystals in CB-CQDs and N-CQDs was poor because of broadening in the interlayer spacing (d) of the CB-CQDs, which was calculated to be 0.38 nm as compared to graphite interlayer spacing (d = 0.33 nm), while FA-functionalized NCQDs improved their crystallinity due to the chemical structure of FA (Figure 3). Meaning that FA was used to facilitate their delivery into a closer region for cancer location. Additionally, it can also support the crystallinity of CQDs.

Sulforhodamine B (SRB) is an aminoxanthene molecule that can penetrate cellular components because of its simple dissolution in a water-forming protein complex in the presence of acetic media. Here, SRB was used to investigate the anti-cancer activity of CB-CQDs, N-CQDs, and FA-NCQDs incubated with A549 cells. The cell viability of CQDs showed low toxicity at a concentration of (1000 μg/mL) (91 ± 1.3%), while cell viability was significantly reduced after their exposure to N-CQDs and FA-NCQDs (12.5 ± 0.14% and 10 ± 0.8% *p* < 0.001), respectively, for 72 h (Figure 4 and Figure 5).

Similarly, cell viability was inhibited significantly after treatment with ethylene diamine alone (1.5 ± 0.2%., *p* < 0.001). This confirms that the toxicity of NCQDs and FA-NCQDs was mainly caused by the presence of ethylene-diamine functionalized CQDs. For this reason, N-CQDs were chosen to be functionalized by FA as a targeted molecule to concentrate the toxicity in the location of the cancer region. FA-conjugated NCQDs let them be taken up by FA-receptor-positive cancer cells, which makes them a new biocompatible probe to distinguish FA-receptor-positive cancer cells from normal cells in biological imaging and cancer diagnosis [57].

In order to explore the morphological alterations associated with apoptotic, necrotic, nuclear fragmentation, and cell membrane damage, A549 cells were exposed separately to 300 µL and 800 µL of (CB-CQDs, N-CQDs, and FA-NCQDs) for different incubating times (24, 48, and 72 h). Cells were mostly observed to be yellow in color in the early apoptotic stage, while in the late stage, apoptotic cells were marked by an orange-red fluorescence stain. (Figure 9). This differentiation was obtained by the ability of acridine orange to penetrate normal and damaged cells, while ethidium bromide can penetrate only damaged cells and then react to their DNA, showing an orange-red color under fluorescence microscopy.

Therefore, the morphology of the nuclear structure of A549 cells was studied by using DAPI stain as a specific nuclear stain. The result showed that cells exposed to N-CQDs or FA-NCQDs showed nuclear condensation and nuclear fragmentation compared to CB-CQDs or untreated cells (control) (Figure 10 and Figure 11). The result indicates that N-CQDs and FA-NCQDs have the ability to cause nuclear damage to cancer cells.

In addition to that, propidium iodide was used to provide a clear understanding of cell and nuclear damage. Propidium iodide, as a selective molecule, cannot cross the intact plasma membrane of non-damaged cells and, therefore, will only be present in the DNA of cells where the plasma membrane has been permeabilized.

In the current study, A549 cells exhibited red fluorescence color after their exposure to N-CQDs and FA-NCQDs compared to CB-CQDs treated cells or untreated cells (control) (Figure 14). These results confirm the cytotoxicity of N-CQDs and FA-CQDs against A549 cells.

To identify the percentage of cells in early apoptosis, late apoptosis, and necrosis, annexin V-FITC was used in the population of flow cytometry gates. The percentage of late apoptotic cells exposed to N-CQDs for 48 h increased significantly to (19.7 ± 0.03%, *p* < 0.05) compared to untreated cells (4.6 ± 0.02%). Meanwhile, necrosis also increased in treated cells to (27.6 ± 0.06%, *p* < 0.05) compared with untreated cells (83.5 ± 0.02%). We, therefore, evaluated the cell cycle in A549 cells after their exposure to N-CQDs for 48 h. The population of cells in the S phase significantly decreased from 29 ± 0.5%, to (14 ± 0.9%, *p* < 0.001) % upon NCQDs (300 µL) exposure for 48 h (Figure 16). A cell cycle study reveals that N-CQDs cause apoptosis.

The current study revealed that induction of apoptosis by N-CQDs and FA-NCQDs was accompanied by an increase in the activation of caspase-3. The level of caspase-3 was evaluated by Elisa kit, and the result showed a significant increase in its level by (2.67 ± 0.2 ng/mL) and (3.43 ± 0.05 ng/mL) in N-CQDs and FA-NCQDs, respectively, compared to untreated cells (0.34 ± 0.04 ng/mL). By the activation of caspase-3, many proteins will be cleaved proteolytically to initiate and progress the programmed cell death.

## 4. Material and Methods

### 4.1. Chemicals

Chicken blood was collected, dried in an oven at 60 °C for 10 h, and then crushed using a mortar. Ethylene-diamine (EDA), Folic acid (FA), Acridine orange, Ethidium bromide (AO/EtBr) stain, 4,6-Diamidino-2-phenylindole dihydrochloride (DAPI) stain, Tween 20, Crystal violet stain, Propidium iodide (PI) stain, and Paraformaldehyde were purchased from Sigma-Aldrich, St. Louis, MO, USA and used as received without further purification. Aqueous solutions were prepared with double distilled water (DDW).

### 4.2. Fabrication and Functionalization of CB-CQDs with Nitrogen Dopant (Nitrogen-Doped CQDs)

The N-CQDs were synthesized using chicken blood as a carbon source and ethylene diamine as a nitrogen source through a hydrothermal process. First, chicken blood (2 g) and ethylene diamine (2 mL) were dissolved in 40 mL of deionized water and placed in a 50 mL glass beaker under vigorous stirring for 15 min. Then, the mixture was transferred into a 75 mL Teflon-lined stainless-steel autoclave at 180 °C for 6 h. After the reaction finished, the suspension was cooled down to room temperature, and a dark brown solution was obtained. The resultant suspension was centrifuged using a cooling centrifuge at 10,000 rpm for 30 min under 10 °C then the solution was purified in a dialysis bag for 48 h (retained molecular weight: (10,000–14,000 Da)). The distilled water was changed every 24 h for the removal of impurities [58].

Afterward, 5 mg folic acid was dissolved in 1mL aqueous solution of sodium hydroxide (1N) and then was dropped into 20 mL phosphate buffer suspension (pH 7.4) with 20 mg of N-CQDs under vigorous oscillation for 30 min (Figure 1). The collected products were centrifuged at 16,000 rpm for 20 min and purified in a dialysis bag for 48 h (retained molecular weight: (10,000–14,000 Da)). The distilled water was changed every 24 h for the removal of impurities. After that, they were lyophilized [59].

### 4.3. Characterization

The characterization of CB-CQDs, N-CQDs, and FA- NCQDs was assessed by UV–vis spectroscopy (Shimadzu UV-2450 double-beam absorbance Spectrophotometer, Tokyo, Japan), Fluorescence spectroscopy, Fourier Transform Infrared Spectroscopy (JASCO FT/IR-6300, JASCO, Tokyo, Japan, model no. AUP1200343), Transmission Electron Microscopy (TEM 200 kV, JEOL-JEM-2100, Tokyo, Japan), X-ray diffraction Shimadzu (XRD 6100, Tokyo, Japan), and Zeta Potential (Brookhaven Instruments, New York, NY, USA).

#### 4.3.1. UV–Vis Spectroscopy

The absorbance of CB-CQDs, N-CQDs, Folic acid, and FA-NCQDs was measured using a Shimadzu UV-2450 double-beam absorbance Spectrophotometer. A total of 500 μL of fabricated nanoparticles was diluted into 4 mL using distilled water and then measured at a range of 200–800 nm.

#### 4.3.2. Fluorescence Spectroscopy

The Photoluminescence emission spectra of CB-CQDs and N-CQDs were measured at room temperature using a spectro-fluorometer Shimadzu RF5301PC. The same sample used in the absorbance measurement was measured with the same conditions, as 500 μL of fabricated nanoparticles was diluted into 4 mL by using distilled water, then measured at the range 200–800 nm. The time-resolved fluorescence measurements were recorded with a picosecond time-correlated single-photon count (Pico Quant, Fluo-Time 200 spectrometer).

#### 4.3.3. Fourier Transform Infrared Spectroscopy (FTIR)

FTIR experiments were carried out using a JASCO Fourier Transform Infrared Spectrometer (FT/IR-6300) to detect the surface molecular structures in the range of 500–4000 cm^−1^ using the KBR pellet method. For all of the tests, at least three scans were recorded on different regions of the samples, and representative spectra were analyzed.

#### 4.3.4. Transmission Electron Microscopy (TEM)

A total of 10 μL of nanoparticles suspension was deposited on the copper grid and air-dried before measurement. Copper grids sputtered with carbon films were used to support the sample. High-resolution TEM images of nanoparticles were analyzed by (JEOL JEM 2100) operating at a voltage of 200 kV, coupled with a GATAN camera ORIUS SC600 with a resolution of 7 megapixels. The GATAN camera was controlled by Digital Micrograph.

#### 4.3.5. X-ray Diffraction

Each sample was dropped separately onto a clean substrate of X-ray. The measurements were performed by Cu-Kα radiation (λ = 1.5405° A) by Powder X-ray diffraction patterns using a Shimadzu (XRD 6100) diffractometer to measure the crystallite pattern and phase structure.

#### 4.3.6. Zeta Potential

The electrophoretic mobility of samples was determined by photon correlation spectroscopy using a Zeta Nano Sizer (Brookhaven Instruments). All measurements were performed at 25 °C. Five following measurements were taken for analysis.

#### 4.3.7. Energy-Dispersive X-ray Spectroscopy (EDX)

Elemental analysis was performed using the silicon-drift EDX detector (energy resolution about 129 eV or better) with the analysis condition of WD 10 mm and voltage 20 KV.

### 4.4. Cellular Experiments (In Vitro Analysis)

#### 4.4.1. Cell Culture

Non-small lung cancer A-549 cell lines were maintained in DMEM (Dulbecco’s Modified Eagle Medium) containing (100 µg/mL) of streptomycin, (100 µg/mL) of penicillin, and (10%) of heat-inactivated fetal bovine serum, with 5% CO_2_ atmosphere at 37 °C [60].

#### 4.4.2. Cell Cytotoxicity Assay

Sulphorhordamine (SRB) assay was assessed to investigate in vitro cytotoxicity of the prepared CB-CQDs, Ethylene-Diamine (EDA), and N-doped CQDs. A-549 (5 × 10³ cells) were cultured in 96 multi-well plates and incubated in complete media for 24 h (37 °C, 5% CO_2_). Then, cells were treated with 100 μL media containing CB-CQDs or N-CQDs at different concentrations (0.1, 1, 10, 100, 1000 μg/mL). After 72 h of exposure, cells were fixed using 150 μL of 10% Trichloroacetic acid (TCA) and were incubated at 4 °C for 1 h. The TCA solution was removed, and the cells were washed 5 times with distilled water. Then, 70 μL SRB solution (0.4% w/v) was added and incubated in a dark place at room temperature for 10 min. Plates were washed 3 times with 1% glacial acetic acid and then were allowed to air-dry overnight. After that, 150 μL of Tris-EDTA (10 mM) was added to dissolve the protein-bound SRB stain; the absorbance was then measured at 540 nm using a BMG LABTECH®-FLUOstar Omega microplate reader (Ortenberg, Germany). The cell viability (%) was calculated from the following equation: (Absorbance of treated group/Absorbance of the control group) × 100% [61,62].

#### 4.4.3. Cellular Uptake

Non-small lung cancer A-549 cell lines (5 × 10³ cells) were cultured in 24 multi-wells and incubated overnight; a new culture medium containing 300 µL of the prepared CB-CQDs, N-CQDs and FA-NCQDs was added per well and incubated for 24 h. Then, cells were washed three times with PBS pH 7.2. Cells were further fixed by using 0.4% paraformaldehyde. Cellular uptake was visualized by red (TRITC), green (FITC), and blue (DAPI) channels of inverted fluorescence microscopy (LEICA). Finally, images were captured using a digital camera [63,64].

### 4.5. Cellular Morphological Alterations

#### 4.5.1. Acridine Orange/Ethidium Bromide Dual Staining Assay

Dual AO/EB staining was used to detect morphological evidence of apoptosis. A549 cells were cultured in a 24-multi-well plate and were treated with (300 µL or 800 µL) of 1 mg/mL CB-CQDs, N-CQDs, and FA-NCQDs separately for 24 h, 48 h, and 72 h. Cells were washed with PBS (pH 7.2), then fixed using 0.4% paraformaldehyde. Cells were finally stained by mixing a concentration of AO/EB (2 mg/mL) for each one in PBS (PH 7.2). After that, cells were examined under a fluorescence microscope (LEICA) at 20× magnification [65].

#### 4.5.2. DAPI Staining

A549 cells were cultured in a 24-multi-well plate and treated using (300 µL or 800 µL) of 1 mg/mL CB-CQDs, N-CQDs, and FA-NCQDs separately for 24 h and 48 h. Cells were then washed with PBS (pH 7.2) and fixed using 0.4% paraformaldehyde. Cells were further permeabilized using 0.1% Triton-X 100 in PBS. Cells were finally stained using 50 µL of 4′,6-diamidino-2-phenylindole (DAPI). Excess dye was removed by washing with PBS, and images of cells were captured under a fluorescence microscope (LEICA) at 20× magnification [66,67].

#### 4.5.3. Crystal Violet Staining

A549 cells were cultured in a 24-multi-well plate and then treated using (300 µL or 800 µL) of 1 mg/mL CB-CQDs, N-CQDs, and FA-NCQDs separately for 24 h, 48 h, and 72 h. Cells were washed with PBS (pH 7.2) and fixed with 0.4% paraformaldehyde; these cells were further washed again with PBS (pH 7.2) and finally stained with crystal violet in PBS (PH = 7.2). After that, cells were visualized under a fluorescence microscope (LEICA) at 20× magnification [68].

#### 4.5.4. Propidium Iodide (PI) Staining

A549 cells were cultured in a 24-well plate and treated using (300 µL or 800 µL) of 1 mg/mL CB-CQDs, N-CQDs, and FA-NCQDs separately for 24 h and 48 h. Cells were washed with PBS (pH 7.2) and fixed with 0.4% paraformaldehyde; these cells were washed again with PBS (pH 7.2) and finally stained with PI diluted with PBS (PH = 7.2). After that, cells were visualized under a fluorescence microscope (LEICA) at 20× magnification [69].

### 4.6. Flow Cytometry Analysis

#### 4.6.1. Annexin V-FITC Apoptosis Detection Assay

The necrotic and apoptotic cells were determined using an Annexin V-FITC apoptosis detection kit (Abcam Inc., Cambridge Science Park, Cambridge, UK) coupled with 2 fluorescent channels flow cytometry. After incubation of A549 cells with 300 μg/mL of N-CQDs for 48 h, cells were collected by trypsinization and washed twice with ice-cold PBS (pH 7.4). Cells were further incubated in a dark place with 0.5 mL of Annexin V-FITC/PI solution for 30 min at room temperature according to manufacturer protocol. After staining, cells were injected via ACEA Novocyte™ flow cytometer (ACEA Biosciences Inc., San Diego, CA, USA) and analyzed for FITC and PI fluorescent signals using FL1 and FL2 signal detector, respectively (λex/em 488/530 nm for FITC and λex/em 535/617 nm for PI). For each sample, 12,000 events were acquired, and positive FITC and/or PI cells were quantified by quadrant analysis and calculated using ACEA NovoExpress™ software (ACEA Biosciences Inc., San Diego, CA, USA) [70].

#### 4.6.2. Cell Cycle Analysis by Flow Cytometry

After incubation of A549 cells for 48 h at 37 °C (5% of CO_2_), Cultured DMEM was changed to a fresh one and was treated with 300 µL of N-CQDs. The untreated and treated A549 cells were collected by trypsinization and then washed twice with PBS (pH 7.2). Cells were further fixed using 60% ice-cold ethanol and stored at 4 °C for 1 h. Fixed cells were washed again with PBS (PH 7.2) and re-suspended in 1ml of PBS containing 50 μg/mL RNase A and 10 μg/mL of propidium iodide (PI) and incubated in a dark place at 37 °C for 20 min. Cells were analyzed for DNA contents by flow cytometry analysis using FL2 (λex/em 535/617 nm) signal detector (ACEA Novo cyte™ flow cytometer, ACEA Biosciences Inc., San Diego, CA, USA). For each sample, 12,000 events were acquired. Cell cycle distribution was calculated using ACEA Novo Express™ software (ACEA Biosciences Inc., San Diego, CA, USA) [71,72,73].

### 4.7. ELISA Caspase Detection

Enzyme-linked immunosorbent assay was used to detect caspase level using a Caspase-3 Activity Kit (C1115; Beyotime Institute of Biotechnology, Haimen, China). A549 cells were exposed to (300 uL) N-CQDs and FA-NCQDs (1mg/mL) for 48 h. Then, cells were lysed and incubated with 2 mM Ac-DEVD-pNA at 37 °C for 4 h. Samples were then measured by spectrophotometry at 450 nm. The analysis procedure was detailed in the manufacturer’s protocol.

### 4.8. Biostatistics Analysis

The results were expressed as mean ± standard error of mean (SEM). Data were analyzed by Sigma Plot Software 12.1 using one-way analysis of variance (ANOVA), followed by Duncan’s test for comparison between different treatment groups. The data are shown as * *p* < 0.05 ** *p* < 0.01 and *** *p* < 0.001. The data are representative of at least three independent experiments.

## 5. Conclusions

In the current work, CB-CQDs emitting blue fluorescence were prepared from chicken blood, which exhibited excellent optical properties and could be used as fluorescent probes for bio-imaging. The physicochemical and photochemical properties of CB-CQDs can be enhanced by chemical doping with heteroatoms (nitrogen). N-CQDs have been successfully synthesized by facile hydrothermal treatment of chicken blood and ethylene diamine. The synthesized N-CQDs exhibited excellent water solubility, stability against an ionic environment, and optical properties. XRD confirmed the amorphous nature of CB-CQDs and N-CQDs. Nitrogen doping was confirmed by EDS elemental composition analysis. HRTEM images showed quasi-spherical morphology without agglomeration. Functionalization of N-CQDs with folic acid was prepared after N-CQDs synthesis. CB-CQDs, N-CQDs, and FA-CQDs were used as fluorescent probes for tumor cell imaging, and cytotoxic analysis was performed to observe the effect of CB-CQDs and N-CQDs in vitro using A549 lung cell lines. It has been observed that CB-CQDs have lower toxicity compared with N-CQDs at high concentrations (1 mg/mL) due to the toxicity of ethylene-diamine added during N-CQDs synthesis. FA has been functionalized on the surface of N-CQDs for targeting therapy, as FA has receptors on the surface of tumor cells, increasing their absorption by these cells and also increasing their effect for therapy. As a result, we can use CB-CQDs for cellular bio-imaging due to their lower toxicity, while N-CQDs and FA-NCQDs can be used for cancer therapy in addition to cellular bio-imaging. The toxic effect of N-CQDs was confirmed by staining (AO/EtBr, DAPI, Crystal violet, and PI), flow cytometry (apoptosis and cell cycle arrest), and ELISA for detecting caspase-3 protein activation.

## Data Availability

Data available in a publicly accessible repository.

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
