# Peer review of "Fighting Non-Small Lung Cancer Cells Using Optimal Functionalization of Targeted Carbon Quantum Dots Derived from Natural Sources Might Provide Potential Therapeutic and Cancer Bio Image Strategies"

_ijms, 2022, doi:10.3390/ijms232113283_

Round 1

Reviewer 1 Report

The authors have reported on an investigation of using natural source-derived carbon quantum dots for the possible treatment of non-small cell lung cancer.  The authors have used a commendable number of recent references and results in clear presentation of figures in the manuscript. The reviewer believes the manuscript is suitable with certain improvements in use of the language.

Author Response

We  would like to take this opportunity to thank reviewer-1 for reviewing our manuscript. The time and effort that you have dedicated to revising our submitted manuscript and providing insightful suggestions and feedback are greatly appreciated. English language was revised thoroughly over all the text.

Reviewer 2 Report

This manuscript by El-borsly et al. describes the development of CDs from natural source for antitumor therapy and bioimaging. 

-The authors should indicate the amount of CDs they obtained from 2g chicken blood and 4 ml ethylene diamine.

-The description of folic acid conjugation with CDs do not mention any dehydrating/activating reagent. Please check. It is unlikely that an amide bond can form under such conditions and the authors should at best obtain a FA-CD salt, which raises concerns as the CDs used in this work are highly negatively charged!

-P4, L125: "The result showed that the cytotoxic activity for CB-CQDs EDA, N-CQDs was detected at (IC50 125 >1000μg/mL, IC50 :95 μg/ml, IC50:304μg/mL) respectively [35,36]." Did the authors determine IC50 for FA-N-CQDs? This is definitely necessary for claiming any FA targeting effect.

-Please indicate the cell line used in legend of figure 5.

-2.2.2. Fluorescence imaging of A549 tumor cells: It is not clear from fig. 6 that decoration of NCQDs with folic acid had any effect on CD internalization in cells. Please discuss.

-P77, L151: " These findings revealed that N-CQDs and FA-NCQDs have the ability to cause apoptosis in A549 cells (Fig.7) ". Considering fig. 7, N-CQDs appear much more efficient to cause apoptosis than FA-NCQDs, revealing that FA targeting is counterproductive. Please discuss.

-Statistic significance in fig. 11 seems doubtful (when comparing N-CQDs to FA-N-CQDs). Please check.

-What are the authors' conclusions from paragraph 2.3.4? What is CB-CQDs in fig.12B?

-Paragraph 2.4.1.: Analysis of FA-CQDs does not appear and again poses the question about FA targeting in this work.

-P15, L298: " These results confirm the anti-cancer activity of N-CQDs and FA-CQDs 298 against A549 cells ". Please rephrase (e.g.: These results confirm the cytotoxicity of N-CQDs and FA-CQDs 298 against A549 cells).

General: 

-Manuscript title does mention targeted CQDs when it does not display any convincing result with respect to cancer cell targeting. Manuscript should be improved with respect to this point, or title changed.  

-Careful proofreading is required to remove abundant typos.

-Quality and homogeneity of the illustrations should be improved.

Author Response

Reviewer 2

I would like to take this opportunity to thank reviewer-2 for reviewing our manuscript. The time and effort that you have dedicated to revising our submitted manuscript and providing insightful suggestions and feedback are greatly appreciated.

Question 1: -The authors should indicate the amount of CDs they obtained from 2g chicken blood and 4 ml ethylene diamine.

Response to Question 1: We would like to thank reviewer-2 for his/her great comment. The amount of CQDs obtained from 10 gm of chicken blood was 500 mg and the amount of N-CQDs obtained was 1300 mg which were obtained from 2gm chicken blood and 2ml ethylene diamine.

.

Question 2: -The description of folic acid conjugation with CDs do not mention any dehydrating/activating reagent. Please check. It is unlikely that an amide bond can form under such conditions and the authors should at best obtain a FA-CD salt, which raises concerns as the CDs used in this work are highly negatively charged!

Response to Question 2: We would like to thank reviewer-2 so much for his/her great comment. According to previous literature reviews  and our previous and recent works, carboxylic groups of folic  acid can be activated  by using EDAC/NHS or by using alkaline solution.

In the recent work, 5 mg folic acid was dissolved in 1mL aqueous solution of sodium hydroxide (1N) and then was dropped into 20 mL phosphate buffer suspension (pH 7.4) with 20 mg of N-CQDs  under vigorous oscillation for 30 min (Scheme1).

This result here was related to de protonate carboxyl group of folic acid in alkaline solution. Nevertheless, their remodulation was obtained after interaction with moieties of capsules and they were detected at 288 and 365 nm respectively. This finding was published in our previous work  and other work as follows

  1. Hanafy NAN, Leporatti S, El-Kemary MA. Extraction of chlorophyll and carotenoids loaded into chitosan as potential targeted therapy and bio imaging agents for breast carcinoma. Int J Biol Macromol. 2021 Jul 1;182:1150-1160. doi: 10.1016/j.ijbiomac.2021.03.189.
  2. Mabrouk Zayed MM, Sahyon HA, Hanafy NAN, El-Kemary MA. The Effect of Encapsulated Apigenin Nanoparticles on HePG-2 Cells through Regulation of P53. Pharmaceutics. 2022 May 29;14(6):1160. doi: 10.3390/pharmaceutics14061160.
  3. Song H, Su C, Cui W, Zhu B, Liu L, Chen Z, Zhao L. Folic acid-chitosan conjugated nanoparticles for improving tumor-targeted drug delivery. Biomed Res Int. 2013;2013:723158. doi: 10.1155/2013/723158.

In the recent work, folic acid dissolved in alkaline solution was attached into amino group of ethylene diamine functionalized CQDs. The result was confirmed by using Xray diffraction, FTIR, UV absorbance and zeta potential measurement.

Question 3: -P4, L125: "The result showed that the cytotoxic activity for CB-CQDs EDA, N-CQDs was detected at (IC50 125 >1000μg/mL, IC50 :95 μg/ml, IC50:304μg/mL, ) respectively [35,36]." Did the authors determine IC50 for FA-N-CQDs? This is definitely necessary for claiming any FA targeting effect.

Response to Question 3: We are highly appreciate the valuable comment were done and we would like to thank reviewer-2 so much. We are fully agreeing with reviewer-2. The cytotoxicity of FA-N-CQDs was studied by SRB reagent for 72h. Cytotoxicity of FA-NCQDs  was provided as (96±0.3%, 94±0.5%, 93±1.5%, 83±1.2%, 10±0.8%) after their incubation to serial concentrations (0.1, 1, 10, 100, 1000 μg/mL) respectively. The result was added to Fig 4 and 5.

Question 4: -Please indicate the cell line used in legend of figure 5.

Response to Question 4: We would like to thank reviewer-2 so much. Non-small lung cancer  (A549 cell line ) was used to estimate IC50 of CB-CQDs, EDA and N-CQDs

Question 5: -2.2.2. Fluorescence imaging of A549 tumor cells: It is not clear from fig. 6 that decoration of NCQDs with folic acid had any effect on CD internalization in cells. Please discuss.

Response to Question 5: We would like to thank reviewer-2 so much for his/her great comment. Folic acid functionalized N-CQDs led to  increase  cellular internalization of FA-NCQDs in A549 cells. The result was demonstrated by fluorescence intensity of accumulated FA-NCQDs. Since, the emission of fluorescence in channel blue and yellow was so intensity in A549 cells exposed to FA-NCQDs compared to those cells exposed to NCQDs or even those exposed to CQDs alone. 

Fluorescence intensity was measured by using image J program and the corrected total cell fluorescence (CTCF) was detected. Since, the fluorescence emission has good intensity in cells exposed to FA-NCQDs. The reason is mainly attributed to accumulate nanoparticles in perinuclear region of cancer cells. This accumulation is due to presence FA that facilitate their cellular uptake Fig 6(A;B and C).  

Question 6: -P77, L151: " These findings revealed that N-CQDs and FA-NCQDs have the ability to cause apoptosis in A549 cells (Fig.7) ". Considering fig. 7, N-CQDs appear much more efficient to cause apoptosis than FA-NCQDs, revealing that FA targeting is counterproductive. Please discuss.

Response to Question 6: We would like to thank reviewer-2 so much for his/her great comment.

In Fig7, both N-CQDs and FA-NCQDs have the ability to cause apoptosis in A549 cells. In addition to that, many A549 cells in section of FA-NCQDs had undergone to cellular degradation and necrosis. In Fig7  FA-NCQDs (k) was sorry changed into more finding one. We ask you please to accept our apologize for such this mistake.

Question 7: -Statistical significance in fig. 11 seems doubtful (when comparing N-CQDs to FA-N-CQDs). Please check.

Response to Question 7: We would like to thank reviewer-2 so much for his/her great comment.

The result was checked by two programs; Sigma Plot, version 12 and EXCEL-ANOVA software as follows: Data, data analysis, t-test: Two-sample Assuming unequal variances, finally choose the variables ranges of both treated groups comparing between them according to ( *P<0.05 **P<0.01 and ***P<0.001). 

The results showed the same statistic analysis.

Question 8: -What are the authors' conclusions from paragraph 2.3.4? What is CB-CQDs in fig.12B?

Response to Question 8: we would like to thank reviewer-2 so much for his/her great comment.

PI was used here because it is more often than other nuclear stains because it is economical, stable and a good indicator of cell viability, based on its capacity to exclude dye in living cells. The ability of PI to enter a cell is dependent upon the permeability of the membrane; PI does not stain live or early apoptotic cells due to the presence of an intact plasma membrane. In late apoptotic and necrotic cells, the integrity of the plasma and nuclear membranes decreases, allowing PI to pass through the membranes, intercalate into nucleic acids, and display red fluorescence.

  1. Rieger AM, Nelson KL, Konowalchuk JD, Barreda DR. Modified annexin V/propidium iodide apoptosis assay for accurate assessment of cell death. J Vis Exp. 2011 Apr 24;(50):2597. doi: 10.3791/2597.

What is CB-CQDs in fig.12B?

CB-CQDs is abbreviated to Chicken Blood Carbon Quantum Dots. CB-CQDs.

Question 9: -Paragraph 2.4.1.: Analysis of FA-CQDs does not appear and again poses the question about FA targeting in this work.

Response to Question 9: we would like to thank reviewer-2 so much for his/her great comment. We are full agreeing with reviewer-2 and we are sorry for such this drawback. We expected that apoptosis /necrosis that was detected after exposure of cells to N-CQDs by using flow cytometry,  could provide also signs for the toxicity of FA-NCQDs particularly, many experiments were done to confirm such this expectation such as SRB cytotoxicity, AO/Eb, DAPI, Crystal violet, PI stains and Caspase-3. 

Question 10: -P15, L298: " These results confirm the anti-cancer activity of N-CQDs and FA-CQDs against A549 cells ". Please rephrase (e.g.: These results confirm the cytotoxicity of N-CQDs and FA-CQDs 298 against A549 cells).

 Response to Question 10: We would like to thank reviewer2 so much for his/her great comment. The sentence was revised.

General: 

Question 11: -Manuscript title does mention targeted CQDs when it does not display any convincing result with respect to cancer cell targeting. Manuscript should be improved with respect to this point, or title changed.  

Response to Question 11: We would like to thank reviewer-2 so much for his/her great comment. We would to provide our apologize.

The title was changed according to suggestion of reviewer-2 as follow; Fighting of non-small lung cancer cells by using optimal functionalization of CQDs derived from natural source might provide potential therapeutic and cancer bio image strategies.

Question 12: -Careful proofreading is required to remove abundant typos.

Response to Question 1: We would like to thank reviewer-2 so much for his/her great comment. English format was thoroughly revised over the text and It may be improved NOW.

Question 13: -Quality and homogeneity of the illustrations should be improved.

Response to Question 13: We would like to thank reviewer-2 so much for his/her great comment.

Quality and homogeneity of the illustrations 

Reviewer 3 Report

The authors aimed at Fighting of non-small lung cancer cells by using optimal func- tionalization of targeted CQDs derived from Chicken blood. Despite the research idea is interesting, However, many concerns regarding the experimental work limits the acceptance of this work. The followings are some of major comments

1-      Many Typographical mistakes were observed. The manuscript should undergo English check from a native English Speaker.

2-      The Introduction Section was deducted into only one paragraph. The authors are encouraged to re-write it again with including more relevant information from previous literature.

3-      In results Section

-          Line 68, N-doped CQDs was mentioned without previous explanation. Each abbreviation should be completely defined at its appearance.

-          Line 91, what is the mean of CB-CQDs

4-      In Section 2.2.1.m why the authors did not investigate the cytotoxicity of folic acid conjugated-NCQDs

5-      Section 2.2.2., comparative quantitative evaluation of cellular uptake should be done.

6-      In section 2.3., the authors checked cellular morphological alterations using different methods. However, the authors used different time points i.e. sometimes 24, 48, and 72 h (Figure 7), and in other setting 24 and 48 h (Figures 8 and 9). Also, the authors did not justified the use of two CQDs concentrations (300 and 800μg) throughout the experiments.

7-      In Section 2.4.1. and 2.4.2., The authors detected apoptosis/necrosis   elicited by N-CQDs. Why the authors did not detect apoptosis/necrosis using FA-NCQDs.

8-      Figure 15 C and D, the presented data with labeled differently, while they represent the same groups.

9-      For caspase-3 detection, western blotting data should be presented.

10-  Lines 267-268, the authors claimed that NCQDS are biocompatible and have low cytotoxicity. From where they derived such information. The authors are recommended to check biocompatibility and cytotoxicity of FA-NCQDs should be investigated against normal cells to confirm their concept.

11-  The Discussion Section seems a duplication of results section. In the discussion section the authors have to explain the obtained results not to rephrase it again. The discussion Section should be re-written in a more scientifically-sound way.

12-  Section 4.3. More experimental detail for each characterization method should be included.

13-  In Section 4.5.2. DAPI Staining was mentioned to be conducted at 24, 48, 72 h. While in the Results Section, only the results of 24, and 48 h were represented.

14-  In Section 4.7., It was mentioned that cells were exposed to N-CQDs, while the results showed that the cells were exposed to both N-CQDs and FA-NCQDs!!

15-  Line 444, the authors mentioned that “The synthesized N-CQDs exhibited excellent water solubility ………” From where they derived such conclusion.

16-  Line 586, Please check the order of references

17-  All figures were of very poor quality and some of them were distorted especially figure 7

Author Response

Reviewer 3

The authors aimed at Fighting of non-small lung cancer cells by using optimal functionalization of targeted CQDs derived from Chicken blood. Despite the research idea is interesting, However, many concerns regarding the experimental work limits the acceptance of this work. The followings are some of major comments

We would like to take this opportunity to thank reviewer-3 for his/her reviewing our manuscript. The time and effort that you have dedicated to revising our submitted manuscript and providing insightful suggestions and feedback are greatly appreciated. English language was revised thoroughly over all the text.

Question 1:  Many Typographical mistakes were observed. The manuscript should undergo English check from a native English Speaker.

Response to Question 1: We would like to thank reviewer-3 so much for his/her great comment. English was revised thoroughly on all the text.

Question 2: The Introduction Section was deducted into only one paragraph. The authors are encouraged to re-write it again with including more relevant information from previous literature.

Response to Question 2: We would like to thank reviewer-3 so much for his/her great comment. The introduction was improved according to reviewer suggestion and  new paragraphs  and Refs were added as follows:

In recent years, carbon-based quantum dots (CDs) that are mainly divided into two subgroups, as carbon quantum dots (CQDs) and graphene quantum dots (GQDs) gained extensive considerations in scientific areas. CQDs were first reported in 2004 by purification of single-walled carbon nanotubes [8]. Meanwhile, CQDs derived from natural source have recently received much attention because of their biocompatible and long-lasting fluorophores for a variety of applications [9-11].

CQD nanostructures are synthesized using various techniques such as microwave irradiation, electrochemical oxidation, hydrothermal method, laser ablation, reflux method, ultrasonication, pyrolysis of glycerol, strong acidic and electrochemical oxidation, thermal carbonization of molecules, thermal annealing of barbecue meat (BBQ) char, as well as atmospheric plasma-based synthesis [14&15]. However, this type of synthesis needs many chemical reactions such as oxidation, pyrolysis and carbonization. Green synthesis of CQDs has several advantages such as renewable resources, use of low-cost and non-toxic raw materials, simple operations, and being environment-friendly [16].

In previous literature, many natural materials used as green sources and precursors to produce CQDs such as  jackfruit, apple, orange, cabbage, banana, pear, jujubes, oolong tea, betel leaf,, bamboo leaf and guava leaf. These materials exhibited outstanding properties including  high renewable capability, low cost, high yield, high availability and high biocompatibility [17]. From blood chicken as a low cost effectiveness materials, just one finding was published. Since CQDs were prepared from chicken blood by hydrothermal method that used as biosensor for catalyzing the oxidation of 3, 3′, 5, 5′-tetramethylbenzidine (TMB) in the presence of H2O2 to generate the blue oxidized TMB (ox-TMB) with a strong absorption peak at 652 nm [18].

In our previous work, FA conjugated NPs derived encapsulated cargo molecules into specific cancer location allowing to increase their cellular uptake. Such this folic acid -folate receptors conjugation was used as a mechanism to facilitate cancer delivery and was studied extensively in breast cancer, lung cancer, and hepatocellular carcinoma [22].

Question 3:  In results Section

-          Line 68, N-doped CQDs was mentioned without previous explanation. Each abbreviation should be completely defined at its appearance.

-          Line 91, what is the mean of CB-CQDs

Response to Question 4: We would like to thank reviewer-3 so much for his/her valuable comment.

N-doped CQDs was explained in introduction as follows

-followed by doping of CQDs with nitrogen source using ethylene-diamine (EDA) to enhance the chemical-physical properties of CQDs with hydrothermal carbonization method [19]. This functionalization was termed as nitrogen doped CQDs (NCQDs).

- CB-CQDs was corrected into CQDs

Question 4: In Section 2.2.1.m why the authors did not investigate the cytotoxicity of folic acid conjugated-NCQDs

Response to Question 4: we would like to thank reviewer-3 so much for his/her great comment. We would to provide our apologize. Cytotoxicity of FA-NCQDs was provided.

Cytotoxicity of FA-NCQDs  was provided as (96±0.3%, 94±0.5%, 93±1.5%, 83±1.2%, 10±0.8%) after their incubation to serial concentrations (0.1, 1, 10, 100, 1000 μg/mL) respectively

Question 5: Section 2.2.2., comparative quantitative evaluation of cellular uptake should be done.

Response to Question 5: we would like to thank reviewer-3 so much for his/her great comment. comparative quantitative evaluation of cellular uptake was added.

In section of “ Material and Methods”

Fluorescence images were first converted into grayscale by using  Image J program https://imagej.nih.gov/ij/. Then cell of interest was selected by freeform option. “set measurements” was opened from “Analyze menu” and parameters of area integrated intensity and mean grey value  were activated. Finally, the bottom of measure was used and the result was analysed

The corrected total cell fluorescence (CTCF) was calculated by using this formula CTCF = Integrated Density – (Area of selected cell X Mean fluorescence of background readings).

In section of “Results”.

Fluorescence intensity was measured by using image J program and the corrected total cell fluorescence (CTCF) was detected. Since, the fluorescence emission has good intensity in cells exposed to FA-NCQDs. The reason is mainly attributed to accumulate nanoparticles in perinuclear region of cancer cells. This accumulation is due to presence FA that facilitate their cellular uptake Fig 6(A;B and C).

Question 6:   In section 2.3., the authors checked cellular morphological alterations using different methods. However, the authors used different time points i.e. sometimes 24, 48, and 72 h (Figure 7), and in other setting 24 and 48 h (Figures 8 and 9). Also, the authors did not justified the use of two CQDs concentrations (300 and 800μg) throughout the experiments.

Response to Question 6: We would like to thank reviewer-3 so much for his/her valuable comment.

In Figure 7, AO/ EtBr dual stains  were used  to stain cells after three different incubating times to evaluate the cytotoxicity at each incubating time allowing us to understand cellular interaction and to identify apoptotic and necrotic stage

In Figure 8 and 9: DAPI stain used to identify the alterations in nuclear morphology after exposure of A459 cells to CB-CQDs, N-CQDs and FA-NCQDs. The results observed presence condensed and fragmented chromatins. While, the morphology was changed after 72h because many cells were ruptured. This cause for us, difficult to account the nuclear condensation and fragmentation.  Fig9 represents  the quantification analysis of nuclear morphologies  of Fig.8.

In the current study, IC50 of N-CQDs was 304µg/mL.  For this reason, the experiments were set in two points:

  • 300µg/mL (mostly as IC50) for investigating the functionalized CQDs by using ethylene diamine as a source of nitrogen
  • 6 fold of IC50 was also used (800μg) to identify the potential cytotoxicity of 800μg for each CB-CQDs, N-CQDs and FA-CQDs. This facilitate our hypothesis to use them later in animal model  

Question7:   In Section 2.4.1. and 2.4.2., The authors detected apoptosis/necrosis  elicited by N-CQDs. Why the authors did not detect apoptosis/necrosis using FA-NCQDs.

Response to Question 6: We would like to thank reviewer-3 so much for his/her valuable comment. We are full agreeing with reviewer-3 and we are sorry for such this mistake. We expected that apoptosis /necrosis that was detected after exposure to N-CQDs by using flow cytometry,  could provide also signs for the toxicity of FA-NCQDs particularly, many experiments were done to confirm such this expectation such as SRB cytotoxicity, AO/Eb, DAPI, Crystal violet, PI stains and Caspase-3. 

Question 8:  Figure 15 C and D, the presented data with labeled differently, while they represent the same groups.

Response to Question 6: We would like to thank reviewer-3 so much for his/her valuable comment.

Figure 15 “C” represents the percentage of cells in different cell cycle stages (G1,S and G2) compared to control. This result identifies qualitatively and quantitatively the percentage of increased or decreased each stage compared to the same stage in control.  For this reason, three stages (G1,S and G2) were presented.

While, Figure 15 “D” represents  the significant reduction in the percentage of cells  of just  S phase compared to the same stage in control. Such this analysis provides simple description for data to be more understanding for reader.

Question 9: For caspase-3 detection, western blotting data should be presented.

Response to Question 6: we would like to thank reviewer-3 so much for his/her valuable comment

We would to express our apologize for this condition because western blot apparatus was not working at time in our institute and we used the techniqual procedure used in previous literatures

  1. Abdullah M, Syam AF, Meilany S, Laksono B, Prabu OG, Bekti HS, Indrawati L, Makmun D. The Value of Caspase-3 after the Application of Annona muricata Leaf Extract in COLO-205 Colorectal Cancer Cell Line. Gastroenterol Res Pract. 2017;2017:4357165. doi: 10.1155/2017/4357165.
  2. Saunders PA, Cooper JA, Roodell MM, Schroeder DA, Borchert CJ, Isaacson AL, Schendel MJ, Godfrey KG, Cahill DR, Walz AM, Loegering RT, Gaylord H, Woyno IJ, Kaluyzhny AE, Krzyzek RA, Mortari F, Tsang M, Roff CF. Quantification of active caspase 3 in apoptotic cells. Anal Biochem. 2000 Aug 15;284(1):114-24. doi: 10.1006/abio.2000.4690.
  3. Arkan, E.; Barati, A.; Rahmanpanah ,M.; Hosseinzadeh, L.; Moradi, S.;  Hajialyani, M. Green Synthesis of Carbon Dots Derived from Walnut Oil and an Investigation of Their Cytotoxic and Apoptogenic Activities toward Cancer Cells. Adv Pharm Bull. 2018 Mar;8(1):149-155. doi: 10.15171/apb.2018.018. 

Question 10:  Lines 267-268, the authors claimed that NCQDS are biocompatible and have low cytotoxicity. From where they derived such information. The authors are recommended to check biocompatibility and cytotoxicity of FA-NCQDs should be investigated against normal cells to confirm their concept.

Response to Question 6: We would like to thank reviewer-3 so much for his/her valuable comment. Normal Human Skin Fibroblast cell line(NHSF) were exposed to serial concentrations (0.1, 1, 10, 100, 1000 μg/mL) of CB-CQDs,  N-CQDs and FA-NCQDs. The result was measured spectrophotometry by using SRB reagent after 72h. The growth of NHSF cells showed significant reduction after their exposure to N-CQDs (101±1.7%, 96±0.6%, 95±1.3%, 83±1.4%, 38±0.7%) respectively compared to CB-CQDs (97±0.7%, 95±1.7%, 94±1.5%, 94±0.9%, 93±2.4%). While, the result of FA-NCQDs  showed (101±1.2%, 98±0.6%, 95±0.7%, 92±0.3%, 90±0.9%) respectively compared to CB-CQDs (97±0.7%, 95±1.7%, 94±1.5%, 94±0.9%, 93±2.4%).

The potential inhibitory effect of concentration (1000µg)  exhibited  (38±0.7%) in case N-CQDs while, less reduction was obtained (90±0.9%) and (93±2.4%) in case FA-NCQDs and CB-CQDs respectively. Thereby,  IC50 of  CB-CQDs,  N-CQDs and FA-NCQDs was detected at (IC50 >1000µg/mL, IC50 : 612.15 µg/ml, IC50>1000µg/mL) respectively. The result confirms that FA  functionalized N-CQDs can  improve their delivery for cancer cells and reduce their toxicity on normal cells.

Question 11:  The Discussion Section seems a duplication of results section. In the discussion section the authors have to explain the obtained results not to rephrase it again. The discussion Section should be re-written in a more scientifically-sound way.

Response to Question 6: We would like to thank reviewer-3 so much for his/her valuable comment. The discussion was re-written as was suggested by reviewer-3 and the results were explained

Question 12:  Section 4.3. More experimental detail for each characterization method should be included.

Response to Question 6: we would like to thank reviewer-3 so much for his/her valuable comment. characterization methods were added as follows:

The characterization of CB-CQDs, N-CQDs, and FA- NCQDs was assessed by UV–vis spectroscopy(Shimadzu UV-2450 double-beam absorbance Spectrophotometer), Fluorescence spectroscopy, Fourier Transform Infrared Spectroscopy (JASCO FT/IR-6300), Transmission Electron Microscopy (TEM), X-ray diffraction Shimadzu (XRD 6100), and Zeta Potential (Brookhaven Instruments).

4.3.1. UV–Vis spectroscopy

The absorbance of CB-CQDs, N-CQDs, Folic acid and FA-NCQDs was measured by using a Shimadzu UV-2450 double-beam absorbance Spectrophotometer. 500 μl of fabricated nanoparticles was diluted into 4 ml by using distilled water and then measured at range 200–800 nm.

4.3.2. Fluorescence spectroscopy

The Photoluminescence emission spectra of CB-CQDs and N-CQDs was measured at room temperature by using a spectro-fluorometer Shimadzu RF5301PC. The same sample, used in the absorbance measurement was measured with the same conditions; as 500 μl of fabricated nanoparticles was diluted into 4 ml by using distilled water, then measured at range 200–800 nm. The time-resolved fluorescence measurements were recorded with a picosecond time correlated single-photon count (Pico Quant, Fluo- Time 200 spectrometer).

4.3.3. Fourier Transform Infrared Spectroscopy (FTIR)

FTIR experiments were carried out by using JASCO Fourier Transform Infrared Spectrometer (FT/IR-6300) to detect the surface molecular structures in the range of 500–4000 cm-1 by using KBR pellet method. For all of the tests, at least three scans were recorded on different regions on the samples and representative spectra were analyzed.

4.3.4. Transmission Electron Microscopy (TEM)

10 μL of nanoparticles suspension was deposited on the copper grid and air-dried before measurement. Copper grids sputtered with carbon films were used to support the sample. High-resolution TEM images of nanoparticles were analyzed by (JEOL JEM 2100) operating at a voltage of 200 kV, coupled with a GATAN camera ORIUS SC600 with a resolution of 7 Megapixel. GATAN camera is controlled by Digital Micrograph.

4.3.5. X-ray diffraction

Each sample was dropped separately onto a clean substrate of X-ray. The measurements were performed by Cu-Kα radiation (λ = 1.5405°A) by Powder X-ray diffraction patterns using Shimadzu (XRD 6100) diffractometer to measure the crystallites pattern and phase structure.

4.3.6. Zeta potential

The electrophoretic mobility of samples was determined by photon correlation spectroscopy by using a Zeta Nano Sizer (Brookhaven Instruments). All measurements were performed at 25 °C. Five following measurements were taken for analysis.

4.3.7. Energy-dispersive X-ray spectroscopy (EDX)

Elemental analysis was performed using the silicon – drift EDX detector (energy resolution about 129 eV or better) with the analysis condition of WD 10 mm and voltage 20 KV.

Question 13:  In Section 4.5.2. DAPI Staining was mentioned to be conducted at 24, 48, 72 h. While in the Results Section, only the results of 24, and 48 h were represented.

Response to Question 13: we would like to thank reviewer-3 so much for his/her valuable comment. We would like to provide our apologize for this mistake. The section was corrected.

Question 14:  In Section 4.7., It was mentioned that cells were exposed to N-CQDs, while the results showed that the cells were exposed to both N-CQDs and FA-NCQDs!!

Response to Question 6: we would like to thank reviewer-3 so much for his/her valuable comment. We would like to provide our sorry for this mistake. The section was corrected.

Question 15:  Line 444, the authors mentioned that “The synthesized N-CQDs exhibited excellent water solubility ………” From where they derived such conclusion.

Response to Question 6: we would like to thank reviewer-3 so much for his/her valuable comment.

According to zeta potential measurement of N-CQDs that observed their surface have  negative charge value . This net charge  produces repulsive force among individual NPs and prevents their aggregation or even glomeration.  Additionally, their solubility in aqueous solution was checked in lyophilized N-CQDs. The obtained result showed that they are soluble in aqueous solution.

Question 16:  Line 586, Please check the order of references

Response to Question 16: we would like to thank reviewer-3 so much for his/her valuable comment. The reference was  changed into recent one.

 Brunetti, J.; Falciani, C.; Roscia, G.; Pollini, S.; Bindi, S.; Scali, S.; Arrieta, U.C.; Vallejo, V.G.; Quercini, L.; Ibba, E.; Prato, M.; Rossolini, G.M.; LIop, J.; Bracci, L.; Pini, A. In vitro and in vivo efficacy, toxicity, bio-distribution and resistance selection of a novel antibacterial drug candidate. Scientific reports, 2016, 6, 26077.‏ doi: 10.1038/srep26077.

Question 17:  All figures were of very poor quality and some of them were distorted especially figure 7

Response to Question 17: we would like to thank reviewer-3 so much for his/her valuable comment.

The figures were improved

Round 2

Reviewer 2 Report

Response to Question 1: We would like to thank reviewer-2 for his/her great comment. The amount of CQDs obtained from 10 gm of chicken blood was 500 mg and the amount of N-CQDs obtained was 1300 mg which were obtained from 2gm chicken blood and 2ml ethylene diamine.

Please, provide this clarification in the manuscript.

Response to question 2 is definitely not acceptable for this reviewer. In the papers by Hanafy et al. and Mabrouk Zayed et al., the authors used NHS and a carbodiimide for folic acid conjugation, which indeed is a classical procedure. In the paper by Song et al., they adsorbed folic acid in the anionic form to deacetylated chitosan which is highly cationic. In this case, electrostatic interactions could indeed occur between the two components as they are of opposite charge. At the contrary, in the submitted manuscript, the nanoparticles are negatively charged and, thus, cannot establish electrostatic attraction with the anionic folate salt.  Moreover, CDs are simply mixed with folate sodium salt and the resulting solution is lyophilized. Of course, any analyses the authors made, they can only reveal the presence of folic acid because nothing was removed from the initial mixture. But, in no case, it can be concluded that folic acid is associated to or interacts with the nanoparticles. This probably explains why targeting with folic acid in this work is so much unconvincing.

Response to question 5 in not convincing. Measurement of fluorescence intensity by Image J on confocal images may have several flaws. Only FACS measurements can provide reliable data to support, or not, the authors' conclusions with respect to folic acid-mediated CD internalization.

In conclusion, This reviewer still has considerable doubts over any targeting effect by folic acid in this work and the data provided by the authors to support this conclusion all are discutable.

Author Response

Response to Question 1: We would like to thank reviewer-2 for his/her great comment. The amount of CQDs obtained from 10 gm of chicken blood was 500 mg and the amount of N-CQDs obtained was 1300 mg which were obtained from 2gm chicken blood and 2ml ethylene diamine.

Please, provide this clarification in the manuscript.

We would like to thank reviewer-2 so much for his/her great interest. The clarification was added in result section (characterization) -page3/lines 102 to 104

In the current study, the amount of CQDs obtained from 10 gm of chicken blood after their purification and lyophilization was 500 mg. While, the amount of N-CQDs  that was obtained from 2gm chicken blood and 2ml ethylene diamine was 1300 mg.

Response to question 2 is definitely not acceptable for this reviewer. In the papers by Hanafy et al. and Mabrouk Zayed et al., the authors used NHS and a carbodiimide for folic acid conjugation, which indeed is a classical procedure. In the paper by Song et al., they adsorbed folic acid in the anionic form to deacetylated chitosan which is highly cationic. In this case, electrostatic interactions could indeed occur between the two components as they are of opposite charge. At the contrary, in the submitted manuscript, the nanoparticles are negatively charged and, thus, cannot establish electrostatic attraction with the anionic folate salt.  Moreover, CDs are simply mixed with folate sodium salt and the resulting solution is lyophilized. Of course, any analyses the authors made, they can only reveal the presence of folic acid because nothing was removed from the initial mixture. But, in no case, it can be concluded that folic acid is associated to or interacts with the nanoparticles. This probably explains why targeting with folic acid in this work is so much unconvincing.

First, We are more than happy for allowing us to discuss in deep our work and to clarify any  other misunderstanding or unclear experiment. We are appreciating much your efforts and we would like to allow us discuss your doubts or concerns.

First of all, the purification of FA-NCQDs was done by using dialysis bag for 48h and the un-reacted materials were removed by distilled water. This step was written clearly in line 669/page 19 to eliminate any doubts or even concern. As, already N-CQDs have been  purified in first step in line 662/page19. The protocol was done according to previous studies as follows:

Yuan, C.; Qin, X.; Xu, Y.;Li, X.; Chen, Y.;Shi, R.;Wang.Y. Carbon quantum dots originated from chicken blood as peroxidase mimics for colorimetric detection of biothiols J. Photochem. Photobio. A Chem., 396 (2020), Article 112529

This means that un-reacted folic acid (that was dissolved by alkaline solution) was already removed by using dialysis bag and fresh distilled water was added. We express our sorry because this point was not so clear in text and NOW it was improved.

Response  to “In the papers by Hanafy et al.”

Hanay et al. had used alkaline solution to de-protonate  carboxyl groups of folic acid. Thereby, under ionic stress of alkaline solution, the active FA was reacted to amino group of BSA (Please have  a look  back to our publication, Paragraph  3.2. Fabrication of CHCAH as a targeted therapy page 1152)

  1. Hanafy NAN, Leporatti S, El-Kemary MA. Extraction of chlorophyll and carotenoids loaded into chitosan as potential targeted therapy and bio imaging agents for breast carcinoma. Int J Biol Macromol. 2021 Jul 1;182:1150-1160. doi: 10.1016/j.ijbiomac.2021.03.189.

Response  to “In the papers by Mabrouk Zayed et al.,.”

It was written  clearly in paragraph “2.2. Fabrication of Albumin-Folic Acid” page 3 “Alternatively, NaOH (1 N) can be used to dissolve and activate FA as well.”

  1. Mabrouk Zayed MM, Sahyon HA, Hanafy NAN, El-Kemary MA. The Effect of Encapsulated Apigenin Nanoparticles on HePG-2 Cells through Regulation of P53. Pharmaceutics. 2022 May 29;14(6):1160. doi: 10.3390/pharmaceutics14061160.

Response to “At the contrary, in the submitted manuscript, the nanoparticles are negatively charged and, thus, cannot establish electrostatic attraction with the anionic folate salt.”

Please before I  have answered this question, I would to ask you kindly to have a look to our previous publication

  1. Hanafy NAN, Quarta A, Ferraro MM, Dini L, Nobile C, De Giorgi ML, Carallo S, Citti C, Gaballo A, Cannazza G, Rinaldi R, Giannelli G, Leporatti S. Polymeric Nano-Micelles as Novel Cargo-Carriers for LY2157299 Liver Cancer Cells Delivery. Int J Mol Sci. 2018 Mar 6;19(3):748. doi: 10.3390/ijms19030748.
  2. Essa ML, Elashkar AA, Hanafy NAN, Saied EM, El-Kemary M. Dual targeting nanoparticles based on hyaluronic and folic acids as a promising delivery system of the encapsulated 4-Methylumbelliferone (4-MU) against invasiveness of lung cancer in vivo and in vitro. Int J Biol Macromol. 2022 May 1;206:467-480. doi: 10.1016/j.ijbiomac.2022.02.095. 

The active Folic acid( EDC/NHS) has a negative charge and was reacted and attached to Polyehylene glycol has terminal OH with ester bond reaction.

 Regarding the recent work, ethylene diamine  as a positive charge reacted to protein, nucleic acid and other biological materials in (Chicken Blood). Therefore, the complex was then  carbonized at 180 áµ’C for 6 hours in  a75 mL Teflon-lined stainless-steel autoclave.

The role of ethylene diamine is to  form crosslink network structures  among moieties of the biological materials of blood chicken (proteins, nucleic acid, fats, other materials)  and to increase ratio of nitrogen in the crystals.  During crystal formation, nitrogen was arranged inside the core of crystals and also on the surface.  Besides that,  surface of nanoparticles contains other chemical groups.  In CQDs,  FTIR spectrum confirms that hydroxyl groups (O-H)  and amide  groups (N-H) were mainly distributed on the surface. Since abroad band was located at 3420 cm-1 and 3200 cm-1.  Additionally, carbonyl group also located at band 1667 cm-1. In N-CQDs, deep band was presented at 3410 cm-1 . This band represents mainly the primary amine/O-H. while,  secondary amine (NH2)  with lower frequency  band was clearly shown at 1568 cm-1. Additionally, carbonyl group was also located at 1650 cm-1.  For this reason, zeta potential value was reduced from (-27mV in case CQDs) to (-17 mV in case N-CQDs). That means different chemical groups were distributed on surface of N-CQDs.

In the recent study, active folic acid was reacted  by amide bond to amino group of ethylene diamine.

To confirm this finding, FA-NCQDs spectrum showed strong abroad peaks located at 3584 cm-1 to 3027 cm-1.  That is related to strong intermolecular hydrogen interaction for both O-H/N-H adsorption.

In meanwhile, there is a high frequency band located at 1560 cm-1 that confirms amide bond formation between carboxyl and amino group compared to this band in N-CQDs that have lower frequency . Additionally, p aminobenzoic acid as a one of folic acid components, is mainly shown by very small peaks arranged in 1461 cm-1 to 1318 cm-1

Response to “Moreover, CDs are simply mixed with folate sodium salt and the resulting solution is lyophilized”.

N-CQDs were first formed and purified by using dialysis bag  for 48h and distilled water was refreshed each24h as it was written in manuscript  “ page19/line 662” according to Ref 18

  1. Yuan, C.; Qin, X.; Xu, Y.;Li, X.; Chen, Y.;Shi, R.;Wang.Y. Carbon quantum dots originated from chicken blood as peroxidase mimics for colorimetric detection of biothiols J. Photochem. Photobio. A Chem.,2020. 396 (2020), Article 112529

Additionally, folic acid functionalized N-CQDs was also purified with the same protocol according to Mabrouk Zayed et al.,

  1. Mabrouk Zayed MM, Sahyon HA, Hanafy NAN, El-Kemary MA. The Effect of Encapsulated Apigenin Nanoparticles on HePG-2 Cells through Regulation of P53. Pharmaceutics. 2022; 14(6):1160. https://doi.org/10.3390/pharmaceutics14061160

 For this reason, there is no any concern related to fabrication/purification of N-CQDs or even FA-NCQDs.

Response to “Of course, any analyses the authors made, they can only reveal the presence of folic acid because nothing was removed from the initial mixture.”

The purification step was done and the un-reacted materials (extra folic acid) was removed. FTIR, XRD, UV visible spectrophotometer revealed exactly the right results for formation/functionalization of CQDs. Please, have a look to the modification bands of the chemical structures that were shown in Fig(1 and 3).

Response to “But, in no case, it can be concluded that folic acid is associated to or interacts with the nanoparticles. This probably explains why targeting with folic acid in this work is so much unconvincing”.

To answer this doubt, FA-NCQDs, N-CQDS  and CB-CQDS were used to treat normal human skin fibroblast for 72h incubation.  The result showed that the growth of NHSF cells showed significant reduction after their exposure to N-CQDs (101±1.7%, 96±0.6%, 95±1.3%, 83±1.4%, 38±0.7%) respectively compared to CB-CQDs (97±0.7%, 95±1.7%, 94±1.5%, 94±0.9%, 93±2.4%). While, the result of FA-NCQDs showed (101±1.2%, 98±0.6%, 95±0.7%, 92±0.3%, 90±0.9%) respectively.

IC50 of  CB-CQDs,  N-CQDs and FA-NCQDs was detected at (IC50 >1000µg/mL, IC50 : 612.15 µg/ml, IC50>1000µg/mL) respectively. The  results confirmed that IC50 of FA-NCQDs was similar to that was obtained by CB-CQDS. This means FA was minimized  cytotoxicity of N-CQDS on normal cells because folate receptors are mainly not expressed by normal skin fibroblast cells.

Response to question 5 in not convincing. Measurement of fluorescence intensity by Image J on confocal images may have several flaws. Only FACS measurements can provide reliable data to support, or not, the authors' conclusions with respect to folic acid-mediated CD internalization.

We would like to thank reviewer-2 for his/her great efforts and interest. Thank you so much for your valuable comments.  We are full agreeing with comment of reviewer-2. However, Image J was used extensively to provide measurement for fluorescence intensity and many other aspects. In the current study, the mean of  fluorescence intensity was obtained and the corrected total cell fluorescence (CTCF) was estimated. The methods was performed according to these references

  1. Feuser, P.E., Jacques, A.V., Arévalo, J.M.C. et al. Superparamagnetic poly(methyl methacrylate) nanoparticles surface modified with folic acid presenting cell uptake mediated by endocytosis. J Nanopart Res 18, 104 (2016). https://doi.org/10.1007/s11051-016-3406-1

  1. Shihan MH, Novo SG, Le Marchand SJ, Wang Y, Duncan MK. A simple method for quantitating confocal fluorescent images. Biochem Biophys Rep. 2021 Feb 1;25:100916. doi: 10.1016/j.bbrep.2021.100916

Response to “ In conclusion, This reviewer still has considerable doubts over any targeting effect by folic acid in this work and the data provided by the authors to support this conclusion all are discutable”.

We are highly appreciate your valuable comments that help us to improve our manuscript. Please if there is any doubts/concern, we are more than happy if we can discuss our work more than one time.

In conclusion:  FA was exactly attached by amide bond to N-CQDs and the other un-reacted FA was removed by dialysis bag and the purification was done for 48h with refresh water. The result of FTIR showed exactly the modification that was done in chemical bands (CQDs, N-CQDs and FA-NCQDs).  Normal skin fibroblast exposed to (CB-CQDS, N-CQDS, and FA-NCQDS) and the result showed that FA-NCQDS obtained IC50 >1000µg/mL that is similar to CB-CQDs

Reviewer 3 Report

The authors have extensively modified their manuscript based on reviewer comments. However, some issues have to be corrected before accepting the manuscript

1- Figure legends for Figures 4, 5, 6 and 9 should be checked and corrected.

2- Reference style should be unified throughout the manuscript

3- The manuscript should be checked by a native English speaker

Author Response

We would like to take this opportunity to thank reviewer-3 for reviewing our manuscript. The time and effort that you have dedicated to revising our submitted manuscript and providing insightful suggestions and feedback are greatly appreciated.

The authors have extensively modified their manuscript based on reviewer comments.

However, some issues have to be corrected before accepting the manuscript

Question 1- Figure legends for Figures 4, 5, 6 and 9 should be checked and corrected.

Response to Question 1: We would like to thank reviewer-3 so much for his/her great interest. The legends of Fig(4,5,6,and9) were revised

Question 2- Reference style should be unified throughout the manuscript

Response to Question 2: We would like to thank reviewer-3 so much for his/her great interest. References were revised and are mostly in the same style of the IJMS/MDPI.

Question 3- The manuscript should be checked by a native English speaker

Response to Question 3: We would like to thank reviewer-3 so much for his/her great interest. I am so sorry and please accept my apologize. I have revised manuscript thoroughly and tried strongly to capture and correct any English Typos with my self because here, we donot have official English office in our university.

Round 3

Reviewer 2 Report

Thank you for these improvements.